

# Risk reduction through managed retreat? Investigating enabling conditions and assessing resettlement effects on community resilience in Metro Manila

Hannes Lauer[1], Carmeli M. C. Chaves[2], Evelyn Lorenzo[2], Sonia Islam[2], Jörn Birkmann[1]

[1]Institute of Spatial and Regional Planning (IREUS), University of Stuttgart, Stuttgart, 70569, Germany
[2]School of Urban and Regional Planning (SURP), University of the Philippines, Quezon City, 1101, Philippines

*Correspondence to*: Hannes Lauer (Hannes.lauer@ireus.uni-stuttgart.de)

**Abstract.** Managed retreat, a key strategy in climate change adaptation for areas with high hazard exposure, raises concerns due to its disruptive nature, vulnerability issues and overall risk in the new location. On-site resettlement or near-site retreat

are seen as more appropriate and effective compared to a relocation far from the former place of living, however, these conclusions often refer to only a very limited set of empirical case studies or do not sufficiently consider different context conditions and phases in relocation. Against this background, this paper examines the conditions and factors contributing to community resilience of different resettlement projects in Metro-Manila. In this urban agglomeration reside an estimated 500,000 informal households, with more than 100,000 occupying high-risk areas. In light of the already realized and anticipated

climate change effects, this precarious living situation exposes families, already socio-economically vulnerable, to an increased risk of flooding. The response of the Philippine government to the vexing problem of informal dwellers has been large-scale relocation from coasts, rivers, and creeks to state-owned sites at urban fringes. Whereas only very few resettlement projects could be realized as In-City projects close to the original living space. The study employs a sequential mixed-method approach, integrating a large-scale quantitative household survey and focus group discussions (FGDs) for a robust comparison of

resettlement types. Further, it reveals community-defined enabling conditions for managed retreat as climate change adaptation strategy.

Results indicate minor variations of well-being conditions between In-City and Off-City resettlement, challenging the expected impact of a more urban setting on resilience. Instead, essential prerequisites for resettlement involve reduced hazard exposure, secure tenure and safety from crime. Beyond these essential conditions, social cohesion and institutional support systems

emerge as significant influencers for the successful establishment of well-functioning new settlements. With this findings, the study contributes to the expanding body of literature on managed retreat, offering a comprehensive evaluation based on extensive datasets and providing entry points for the improvement of retreat as a climate change adaptation strategy.



## 1 Introduction

Planned relocation or managed retreat is an increasingly accepted adaptation strategy to mitigate the potential effects of
climate-induced hazards such as flood disasters or sea level rise (Haasnoot et al., 2021; Mach and Siders, 2021; Ferris and Weerasinghe, 2020; Carey, 2020; Greiving et al., 2018; Hino et al., 2017; IPCC, 2014). Often, this entails uprooting entire communities and transferring them to safer locations where families can start anew without the constant threat of disasters and displacement. However, moving families out of harm's way is not synonymous with building their capacity to stem disaster impacts, much less catalysing their recovery from disruptions. Beyond a change in residence, planned relocation has to be an
enabling process for communities (Arnall, 2019).

This holds particular significance in the context of the Philippines and Metro Manila, where resettlement efforts principally target the most vulnerable, namely those hundred thousand residing in informal settlements (Alvarez, 2019; Ballesteros and Egana, 2013). The Philippines has a longstanding tradition of resettlement, primarily associated with slum clearance initiatives and development projects, such as highway construction (Lauer et al., 2021; Ajibade, 2019). More recently, and notably
following the devastating effects of Typhoon Ondoy in 2009 (international name: Ketsana), there have been ambitious plans to resettle the people living in designated danger areas (Alvarez, 2019; Galuszka, 2019; Ballesteros and Egana, 2013) as integral part of disaster risk reduction (DRR) strategies and intentions to flood-proof the city (Ajibade, 2019; Alvarez, 2019). One concern arises in this context, namely that the declaration of the danger areas is solely based on the national water code from 1976 and not on profound risk understandings. Thus, danger areas are not necessarily areas with a distinct flood
probability or taking into account vulnerable and sensitive elements. Nor do they consider future climate change impacts and associated risks. And these risks are significant, when regarding, for example, that storm surges in the wake of typhoons are already threatening vast areas of urbanized Metro Manila coastline and kilometres of hinterland (Lapidez et al., 2015; Tablazon et al., 2015) and that rising sea levels are accelerated by rapid land subsidence in the region (Cao et al., 2021; Jevrejeva et al., 2016; Rodolfo and Siringan, 2006). The second concern in the context of danger zone resettlement is that the default strategy
of resettlement in Metro Manila has ever been large-scale Off-City settlements where the so-called beneficiaries are relocated in a top-down manner to the periphery of the National Capital Region or even to rural regions in neighbouring provinces (Galuszka, 2020; Jensen et al., 2020; Ballesteros and Egana, 2013). On the other hand, only few examples of In-City settlements, where individuals are resettled within the same municipality in rather close proximity to their original living space exist. Therefore critics argue that current resettlement activities are a new form of eviction in the name of saving lives (Alvarez,
2019) and leading to significant disruption of lives of the resettled people (Mateo, 2022; Jensen et al., 2020; Tadgell et al., 2017).

While managed retreat gains global traction as a climate change adaptation and risk reduction strategy, and resettlement has become a central component of urban development in Metro Manila, there is a noticeable gap in the evaluation of resettlement practices. Moreover, the expanding literature on managed retreat also lacks robust comparisons and evaluations using extensive
datasets (Haasnoot et al., 2021; Greiving et al., 2018), indicating that the current and long-term impacts of managed retreat





remain inadequately assessed in terms of success or effectiveness (Hoang and Noy, 2020). Whereby the challenge starts already in defining what success should mean for retreat (Ajibade et al., 2022; de Sherbinin et al., 2011). Against this background, this study holds particular significance as it aims to contribute to filling this gap by evaluating and scrutinizing the suitability of retreat as an adaptation strategy and providing insights into the living conditions of resettled communities. What distinguishes

this study is its approach - the success or the effectiveness of resettlement is defined by the resettled communities themselves. Meaning, the communities resettled from flood-affected informal settlements identified elements crucial for their community resilience. These elements are considered as community-defined enabling factors for resilient retreat and serve in this study as categories for the assessment. Apart from the desire to distil these enabling factors, the two research questions based thereon are: 1. Are resettlement practices in the Philippines deemed successful when assessing these community-defined enabling

factors? 2. Are there significant disparities when comparing In-City resettlement projects with their Off-City counterparts?
In pursuit of the research objective and exploration of these research questions, the study employs a comprehensive mixed-methods approach with a large-scale quantitative household survey and focus group discussions (FGDs). The FGDs revealed the community perceptions of what accounts for community resilience in the context of their settlement while the quantitative household survey provided essential data for analysing the defined enabling conditions. Drawing from these first hand

experiences and robust data, the study explores the essential conditions that contribute to the effectiveness of resettlement efforts at the household level.
The following second section sets the stage by delving into the state of the research field of retreat as hazard prevention and climate change adaptation strategy as well as into the historical context and contemporary landscape of resettlement practices in the Philippines. The third section elaborates the applied methodology. It describes the study design and informs about the

conducted survey and the FGDs. Section four provides the results derived from the applied methods, while section five offers a synthesis. The final section six concludes by revisiting the initial research objective and further contextualizes the results within the realm of policy implications.

## 2. Perspectives on managed retreat as an adaptation strategy

### 2.1 Retreat, resettlement in the context of climate change and habitability

Climate change is a significant risk amplifier, impacting both on slow and sudden onset hazards (Birkmann and Lauer, 2022): Sea level rise and salinization are threatening coastal regions, home to hundreds of million (Glavovic et al., 2023; Kulp and Strauss, 2019). Extreme heat endangers human-wellbeing, while extreme rain and storm events are likely to become more frequent and intense (Lee et al., 2023). These changes often impact areas already exposed to severe hazards, raising concerns about habitability. This discussion revolves around which regions will support healthy and sustainable living or even survival

- today but especially in a world projected to be 2 or 3 degrees warmer (Mach and Siders, 2021). Exploring habitability involves understanding potential thresholds in three dimensions: basic human survival, livelihood security and the capacities of societies to manage environmental risk (Horton et al., 2021). Within this context, discussions are increasingly concerned about present



and future migration triggered by environmental and climate change, coupled with societal factors (Sakdapolrak et al., 2023; Szaboova et al., 2023; McLeman et al., 2021; Hauer et al., 2020; Scott et al., 2020; McLeman, 2018; Adams, 2016; Black et

al., 2011). One facet of such migration influenced by climate and environmental change is planned retreat, or resettling exposed populations to safer areas. Similarly, as climate change is amplifying existing hazards and therewith risks, it is also amplifying resettlement activities. Certainly, resettlement is nothing new. It has been a relevant topic before climate change, and would remain critical even in the absence of future climate impacts. This is because it intertwines with ongoing planning discussions involving topics such as urban poverty, the right to the city, gentrification, social housing, equality, rapid urban growth or

disaster risk reduction. However, its significance is set to increase tremendously under conditions of climate change (Haasnoot et al., 2021; Mach and Siders, 2021; Scott et al., 2020;). Government-led or planned resettlement to safeguard populations at risk from disasters aligns with the conception of planned adaptation. Planned adaptation arises from decisions rooted in recognizing changing conditions or anticipated changes, necessitating actions to attain, maintain, or return to a desired state. These actions might involve interventions to prevent, tolerate, spread the loss, or change location. These measures can be

further classified by their function as protect, accommodate, or retreat (IPCC, 2014). However, the choice of any planned adaptation action is value-laden, requiring governments to make prior decisions about what to preserve, alter, or permit to stay its course. Planned adaptation decisions are commonly made at superordinate national levels, and it is thus criticized that government policies and practices might be inadequate to take into account context-specific vulnerability-poverty linkages (Rahman and Hickey, 2019). In most cases, the initial response of planned adaptation is to protect what is valued. If not

possible, the approach is to accommodate and bear certain losses. The last resort is to retreat from a location when spreading the loss is no longer viable. While this argument merits rights-based policy discourse, disaster-prone countries like the Philippines will assert the moral imperative to protect their population and resources from imminent risk as a precaution to prevent or stem future loss and damage.

Besides the conceptual or heuristic examination of planned or managed retreat, it can be assumed that on the practical level, it

builds mainly on existing practices of resettlement. It is unlikely that resettlement practice would dramatically change all of a sudden, solely as hazard risk and climate change are proclaimed as reasons for relocating people. Considering the significant risks associated with resettlement (Ajibade et al., 2022; Dannenberg et al., 2019; Rogers and Wilmsen, 2019; Maldonado et al., 2013; De Wet, 2006; Cernea, 1997; Scudder, 1993) it would pose potential problems when existing resettlement strategies and practices were merely continued or even extended under the guise of risk prevention as managed retreat. This might further

be difficult, as it can be estimated that informal settlers are well sensitized, based on their experience with land interest and development-induced resettlement, that the land they are living on is of value and the actual reasons of their proposed resettlement might be others than their safety (Pérez et al., 2022; Saguin and Alvarez, 2022; Alvarez, 2019). Therefore, it is crucial to scrutinize and evaluate current practices and policies, aiming to identify enabling conditions that facilitate resilient forms of retreat, steering clear of previous or current mistakes being made in resettlement practice.





### 2.2 Resettlement as adaptation in Metro Manila, Philippines

Resettlement in Metro Manila mainly deals with relocating the urban poor, especially those in informal settlements (Jensen et al., 2020). Since the 1960s- and 70s, various policies and programs targeted informality, ranging from public housing to slum upgrading and centralized relocation applying a low-rise housing typology to the outskirts of the urban areas (Lauer et al., 2021; Du and Greiving, 2020; Galuszka, 2020; Ballesteros, 2002). Despite these multiple efforts, coordinated centrally by the National Housing Authority (NHA) since its establishment in 1975, major challenges in the housing sector persist. These include a significant housing backlog, high homelessness rates and a substantial population residing in informal settlements (UN-Habitat, 2023).

Initially, resettlement served as an instrument for slum clearance to address tenure issues or to cater private interests. Later, the focus shifted towards infrastructure development and the rehabilitation of public spaces (Jensen et al., 2020; Delos Reyes and Francisco, 2015). Since the end of the 2000s, environmental and risk concerns created an urgency and motif for relocating families. In 2008, the Supreme Court issued a writ of continuing mandamus instructing different government departments and agencies to clean up and rehabilitate the Manila Bay to a quality of water fit for swimming and other recreational activities. This clean-up initiative was extended to rivers and major tributaries feeding into the Bay. Integral to the rehabilitation effort is the intensified conservation of buffers for waterways in urban areas by removing illegal structures in the 3-meter easement area, which was also named danger area. In the wake of this directive, Typhoon Ondoy wrought unprecedented havoc in Metro Manila and adjacent provinces, which bolstered the clearing operations anew. Resettlement then took a different impetus. By 2011, the newly installed administration initiated a 5-year resettlement program called the Oplan Lumikas para Iwas Kalamidad at Sakit (LIKAS), which aimed to relocate roughly 120,000 informal settler families (ISFs) from the danger areas along major waterways in Metro Manila to other parts of the metropolis and neighbouring provinces (Galuszka, 2020). The government earmarked 50 billion Pesos in funding to produce 120,868 dwelling units for qualified informal settler families within and around Metro Manila to respond to the court order while addressing the long-drawn clamor for housing by a ballooning IS population estimated to be around 1.2 million in NCR alone (World Bank, 2017a). By the end of the program, the Department of the Interior and Local Government (DILG) reported that the program completed around 73,5 % of the targeted housing units (Galuszka, 2019). Over 80 % of these structures were established in Off-City resettlement projects (Galuszka, 2018; World Bank, 2017b), an approach that Ballesteros (2017) described to be less effective in delivering the expected socio-economic benefits. This observation resonates with recent legislative reforms to institutionalize In-City resettlement over Off-City option (Republic of the Philippines, 2017). Despite this articulated push for In-City relocations, practical challenges remain and thus, large Off-City developments are still the default and more easily replicable strategy. Several factors come into play, including the availability of affordable or suitable lands for socialized housing in the Metro, the quantity of families to be relocated or the urgency for fast-processing relocation of informal settlers.

The discussion about Off-City and In-City resettlement is still a controversial and intensively debated topic within the discourse on resettlement, underlining the argument that relocating to Off-City sites may disrupt the existing livelihoods and potentially



leading to a high risk of impoverishment and, in some cases, even a return to informal urban areas (Mateo, 2022; Doberstein et al., 2020; Jensen et al., 2020; Tadgell et al., 2017). This ongoing discourse has gained fresh momentum with the new
administration's ambitious plan to develop 1 million housing units annually until 2028 nationwide. This initiative aims to address the massive housing backlog, estimated at around 6 to 6,5 million, and to ultimately eliminate the number of ISF to zero: "Pambansang Pabahay para sa Pilipino: Zero ISF Program for 2028" (Katigbak and Teodoro, 2023; Bautista, 2022). According to this political initiative, it is estimated that still 500.000 ISF live in Metro Manila, with many of them in the designated danger areas, despite the effort brought forward within the Oplan LIKAS program. Most of their planned
resettlement is nowadays intended to be accomplished through the construction of high-rise In-City housing (Mateo, 2022; Parrocha, 2022). However, the proposed target of 1 million units per year is highly ambitious, significantly surpassing the current output of the entire housing industry by sevenfold, and not even speaking about the financial needs for such program (Jose S. de Guzman, 2023; Katigbak and Teodoro, 2023). Together, the recent efforts to resettle individuals from danger areas, coupled with the ambitious plans to tackle informality, will likely accelerate resettlement activities in the Philippines,
especially in Metro Manila. This direction towards rapid and substantial housing delivery for ISF seems to be leaning more towards a standardized top-down or investor-driven development model. This would contradict the desired co-creation or People's Plan approaches outlined in policy documents and programs such as in Oplan Likas (Galuszka, 2020, 2019). Such alternative resettlement practices would need more diverse housing and program typologies, financing schemes and budget allocation, addressing the needs of those being resettled and giving emphasizing to social preparation before and long-term
post relocation activities (Lauer et al., 2021).

### 3. Materials and Methodology

#### 3.1 Research design

This study follows a sequential multi-method design, where each subsequent step build upon the method and the findings of the preceding one. Along the two years study, the research questions and methods had to be adapted to changing conditions
and gained insights. However, the overall underlying research themes are, as can be seen in Figure 1 to evaluate and scrutinize the suitability of retreat as an adaptation strategy and to compare Off-City with In-City settlements. The point of departure in the initial phase of the research was to investigate differences between the two settlement categories by data of a mainly quantitative household survey. The analysis envisaged an assessment and comparison of the hazard exposure and livelihood situation both, in the current settlement and in contrast to the former location before the resettlement. The hypothesis was, that
In-City settlements show better results as the interruption into resettle's lives is supposed to be smaller. An initial analysis of the data in the analysis phase one uncovered partially surprising results (detailed in section 4), in some instances challenging this hypothesis.



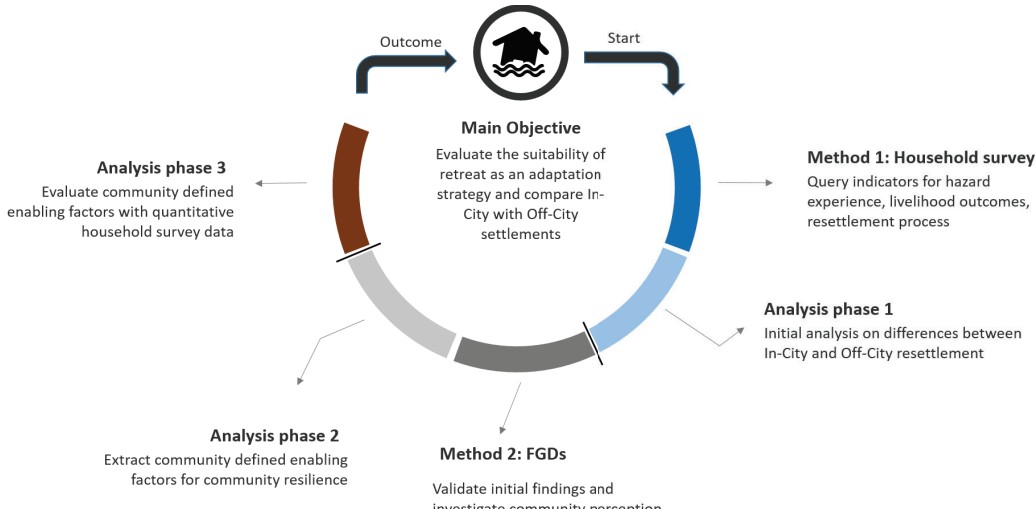

**Figure 1: Research design**

Accordingly, to validate and expand upon these results, the next step within the research process was to employ qualitative methods, engaging with residents of two settlements through FGDs. This qualitative approach excavated the perception of the community on essential elements that contribute to their resilience in the analysis phase two. These community-defined factors were discussed and ranked indicating their importance for the community. They are understood in this research as community defined enabling factors for resilient retreat. In the last step of the research process, the analysis phase three, these enabling

factors were then analysed with the prevailing data from the household survey (method 1) to ultimately deal with the underlying research objective, thus evaluate how the resettlement satisfied these factors. This quantitative evaluation of community defined factors, investigating on retreat effects and successes, is complemented in this research step by narratives from the FGDs.



### 3.2 Study area and resettlement sites

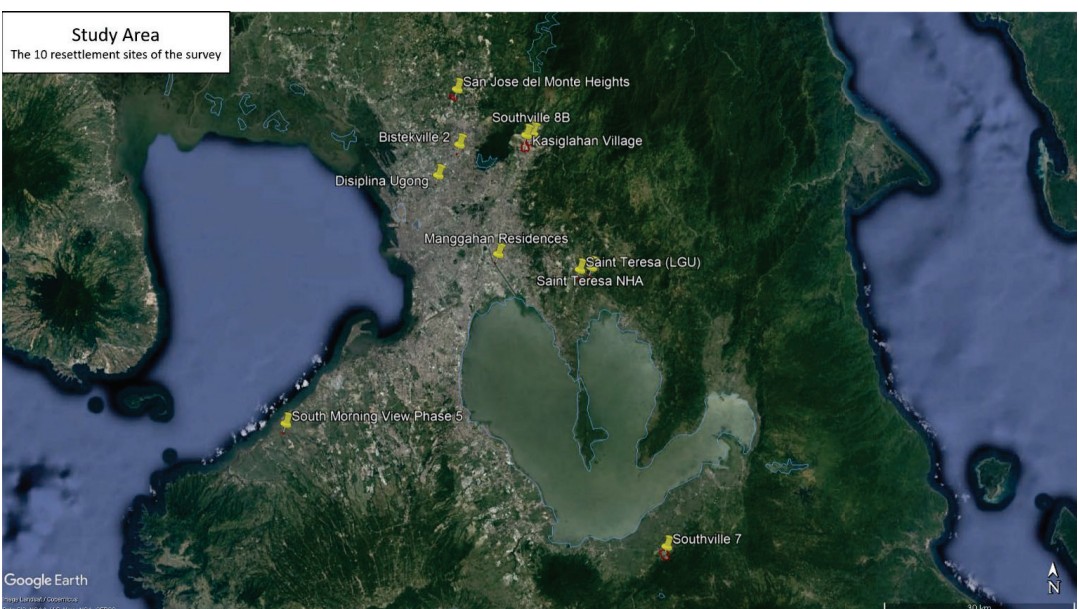


**Figure 2: The 10 research sites, Map data: © Google, SIO, NOAA, US Navy, NGA, GEBCO; Image: Landsat/Copernicus**

The research was conducted in 10 resettlement sites, comprising three In-City settlements situated within the National Capital Region, commonly referred to as Metro Manila. They are located in the City of Pasig, in Valenzuela and in Quezon City. The remaining 7 settlements fall into the category Off-City or peripheral Near-City settlements, located in the provinces of Bulacan,

Rizal, Cavite and Laguna. The selection process is grounded in a pre-established typology of resettlement projects in Metro Manila and surrounding provinces as outlined in prior work (Lauer et al., 2021). Consequently, to ensure a comprehensive overview and encompass the diverse range of resettlement approaches, the study selected settlements exhibiting variations in the four key components: location, program and finance, strategy, participation, and housing types. It is crucial to note that in the political discussion on resettlement in the Philippines, the primary factor distinguishing settlements is their location. This

is mirrored in categorizing of settlements as either an In-City or an Off-City resettlement type. However, even if this broad categorization is also followed in this research, it is essential to recognize that the settlements are divers and can differ from each other even if they are in the same location category. For instance, some Off-City settlements, such as those in Cavite and Laguna, are situated in considerable distance to urban areas, with distances exceeding 40 or even 70 kilometres. On the other hand, certain settlements were originally constructed in rural areas at the fringes of Metro Manila around 20 years ago. But

due to the rapid expansion of urban areas in the last decades, some of these settlements, such as those in Montalban, Rizal, are no longer considered highly peripheral.



### 3.3 Household survey

In the course of the household survey between March and Mai 2022, a total of 1167 households were interviewed by a team of 12 professional enumerators in the local language Tagalog. The results of 30 households had to be excluded from the
analysis as of missing values for important indicators. Accordingly, the final sample size is 1137. The questionnaire consisted of 170 questions, mainly of quantitative nature encompassing Yes/No responses, single- and multiple choice formats as well as Likert Scale questions. These questions addressed both the individual person surveyed and the household she or he lives in. Examples for questions directed to the individual include inquiries about the respondent's educational level or their personal feeling of security. Questions in which the respondent serves as a representative of the household include inquiries about the
monthly household income or whether the household has experienced flooding.

Following the survey, the collected data were digitalized and translated to English. These datasets were then organized within MS Excel and further processed in SPSS, including the necessary coding of questions and variables. The categorization of the settlements with the sample size can be seen in Table 1. The selected sites and the sample reveal that the three settlements categorized as In-City settlements provide 323 respondents which accounts for 28,5 % of the overall sample size of 1137
respondents.

**Table 1: The research settlements and their basic facts**

| Settlement name | Location component | | Main time of resettlement | Inhabitants and sample size | | |
|---|---|---|---|---|---|---|
| | Urban (In-City, Near-City – urban) | Peripheral (Off-City, Near-City peripheral) | Median year of resettlement of the respondents | Approx. number of available housing units | Initial sample size N = 1167 | Final sample size N = 1137 |
| Bistekville 2 | x | | 2015 | 1078 | 107 | 83 |
| Disiplina Village - Ugong | x | | 2014 | 892 | 120 | 120 |
| Manggahan Residences | x | | 2018 | 489 | 120 | 120 |
| Kasiglahan Village | | x | 2000 | 9915 | 155 | 153 |
| San Jose del Monte Heights | | x | 2013 | 6500 | 120 | 119 |
| South Morning View | | x | 2017 | 1180 | 120 | 120 |
| Southville 8b | | x | 2010 | 8280 | 150 | 149 |
| Southville 7 | | x | 2010 | 5000 | 150 | 149 |
| Saint Teresa LGU | | x | 2018 | 250 | 50 | 50 |
| Saint Teresa NHA | | x | 2014 | 270 | 75 | 74 |

The sampling method employed in this survey followed a heterogeneous or flexible purposive multi-stage approach. It involved a combination of different probability sampling techniques tailored to the different settlement types. In smaller
settlements, the research team initiated the process with a systematic sampling, selecting every xth house, and subsequently,



utilized a snowball sampling to achieve the desired sample. In larger settlements, the interviews were conducted in all areas or quarters of the settlement and representatives of the Homeowner Associations (HOAs) who identified the areas with households that were relocated from waterways mostly guided the enumerator team. Additionally, snowball sampling was applied. In the case of In-City settlements consisting of medium-rise buildings, each building block was assigned to a specific

enumerator who was responsible for conducting systematic and then snowball sampling within the designated block to identify resettled households. This sampling method could ensure that most respondents can be understood as having experienced managed retreat activities. Out of the 1137 respondents 988 or 86,9 % mentioned that the reason for their resettlement was that they were living in a danger area.

### 3.4 Focus Group Discussions

The two FGDs took place in October 2022 in the settlements Kasiglahan Village and Saint Therese Housing (NHA), both situated in Rizal province. These two settlements, however, have markedly different characteristics. Kasiglahan Village, located in the municipality of Rodriguez, is a relatively aged settlement, which was established more than 20 years ago. It is a large community with various quarters, named phases, comprising of a total of over 9000 housing units. Today, it finds itself in a rather urbanized area as Rodriguez witnessed urbanization processes in the past decades. Notably, Kasiglahan Village has

faced severe flooding issues during recent typhoons in some phases of the settlement. Saint Theresa Housing, on the other hand, is located in the municipality of Teresa. It is a small settlement with 270 housing units. Although Teresa is in proximity to Antipolo, a rapidly growing city just adjacent to Metro Manila, the settlement itself is situated in a remote, almost rural setting.

Overall, 26 participants provided insights into the community processes, social behaviour, and development conditions they

perceive to shape community resilience. In both settlements, the FGD participants were long-time residents of the sites. They represented different social groups, namely women, youth, community leaders, fathers, people with disabilities, elderly and LGBTQ+. Recordings and two external observers documented the discussions of the FGDs. MS Excel was used by the research team to code the FGD recordings, identifying the commonly occurring words, conditions, and descriptions categorized into the resilience elements.

## 4. Discussion of results

The results section follows the elaborated research design depicted in Figure 1 This implies that firstly, the initial comparison between In-City and Off-City settlements is presented (analysis phase one), followed by an elaboration on the FGDs which distilled the enabling factors (analysis phase two). Finally, the section concludes with the in-depth analysis of the enabling factors by using the quantitative data (analysis phase three).



### 4.1 Initial analysis to compare In-City with Off-City resettlement


The initial analysis of the survey data encompassed the use of descriptive statistics such as exploratory data analysis or frequency distribution for most indicators. An interesting first result is that when comparing the settlements regarding hazard exposure and livelihood outcomes we found that the difference between surveyed settlement categories (such as Off-City versus In-City) was less significant as expected. Meaning that solely minor variations could be found when comparing the 7

Off-City settlements with the 3 In-City settlements. Even for some indicators the Off-City category encompassed better results and thus in some cases the settlement with best results was an Off-City relocation settlement.

Three selected indicators briefly illustrate this observed trend in the data. The first indicator involves a comparison of safety against natural hazards. This is a central measure, considering that the reduction of hazard exposure, particularly minimizing flood risk and the clocking of waterways, served as the rational and justification for clearing the declared danger areas. The

data of the household survey reveals that the major objective of improving hazard security has been largely achieved for the majority of resettled individuals. To be precise, 90,6 % or 1030 out of the 1137 respondents affirmed that their current resettlement offers greater safety against natural hazards compared to their former residence in informal settlements. A mere 1,0 % expressed that their current settlement is less secure, while 8,4 % reported a comparable level of safety their previous situation. Furthermore, when contrasting In-City settlements with Off-City settlement, the data demonstrates that in In-City

settlements the rate of respondents who perceive the settlement as safer against natural hazards is marginally higher with 92,0 % as in their Off-City counterparts.

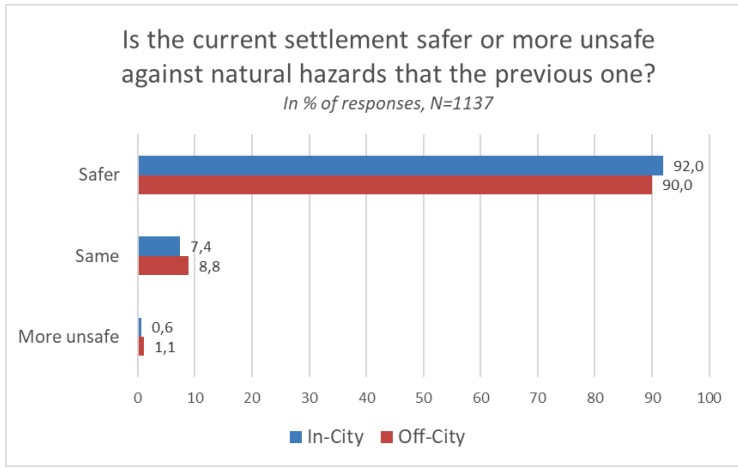

**Figure 3: Comparison of perceived safety against natural hazards**

The second noteworthy indicator is the impact of the resettlement process on the quality of life for families. This indicator

provides a very general yet summarizing view of perceived resettlement outcomes. An overall positive picture emerges for



this indicator, with 44,2 % expressing a positive influence, 26,8 % reporting slightly positive influences, while only 7,4 % mentioned a negative influence, and 16,7 % noted a slightly negative effect. Figure 4 shows that the results slightly favor a more positive picture in In-City settlements, where a combined total of 73,7 % reported positive or slightly positive influences, compared to 69,9 % in Off-City settlements.

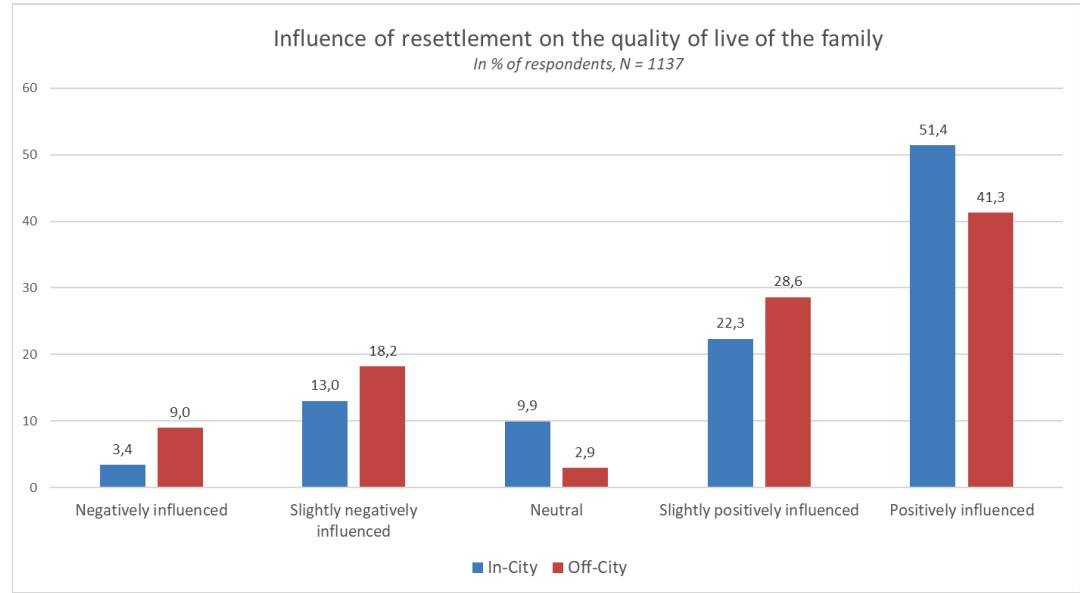

**Figure 4: Comparison of the development of the quality of live**

The third indictor exemplifying the trend of the initial analysis focusses on monthly household income. Income hereby encompassed the total inflow of money from salaries, wages, self-employment, remittances, rent or other sources. The mean estimated monthly income is 16.970 Pesos with a minimum of 500 and a maximum of 133.500 Pesos. When classified into the two settlement types, it becomes evident that in In-City settlements, the mean income is 1.477 Pesos higher, and the median income is 1.600 higher than in Off-City settlements. In terms of currency conversion, this higher amount roughly equates to 25 Euros or 25 US Dollars when fluctuations in exchange rates are not considered. Deeper analysis found that one In-City settlement (Bistekville 2) reported higher income as others with 20197,6 Pesos (median 20.000), significantly more than the settlement with the second highest income, the far-away Off-City settlement South Morning View with 182227,6 Pesos (median 15.000).



### 4.2 Community perception of resilient retreat

As highlighted in section 3.1, the purpose of the FGDs was twofold: to validate the initial survey results and to delve deeper into how resettled communities perceive resilience to flooding given their lived experiences in coping with the impacts of the hazard. Hence, the participants were asked how they themselves think about their community resilience, hereby giving less weight to the academic discourse on resilience but more to their personal definitions and perceptions. Notably, many participants of the FGDs were already aware about the concept resilience or at least had heard about it in the broadest sense of the community's ability to withstand shocks and fortify the quality of life. This basic understanding stems from their collaborations with local authorities or involvement in HOAs and NGOs.

The vivid discussions revealed that resilience as a collective capacity remains highly context-driven. Nevertheless, cross-cutting features suggest similar resilience-enhancing conditions mentioned in the long-established resettlement sites Kasiglahan Village and in the younger resettlement Saint Therese Housing. Based on their own understanding of resilience, as we did not further define it, the participants named and discussed various conditions associated with a resilient community. Among these identified conditions, a safer location and better neighborhood security tops the list of perceived essential elements of community resilience, as 88 % of the 26 participants in the FGDs claimed similar observations. Around 80 % of the participants further mentioned that possessing a legal document for their occupancy made them feel more secure and hopeful about owning their homes.

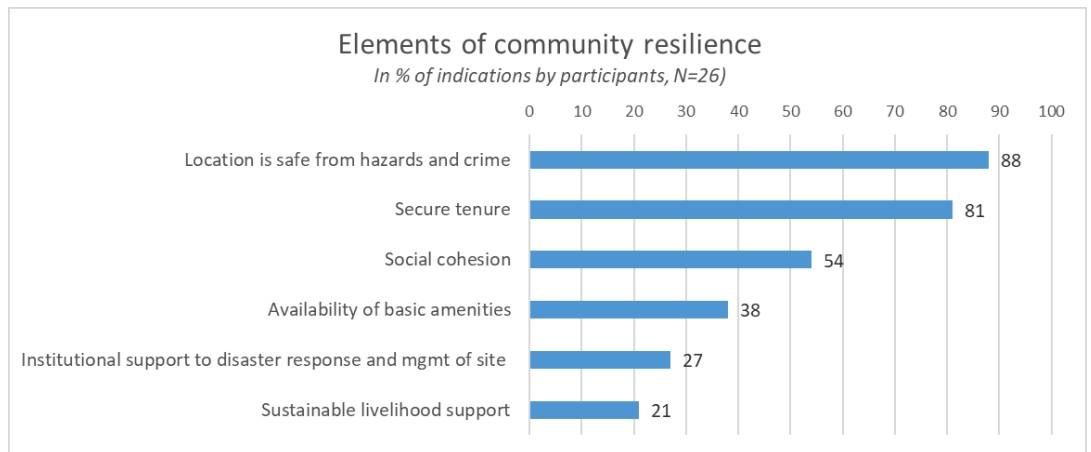

**Figure 5: Perceived essential elements of community resilience ranked by the frequency of the indication as relevant for FGD participants (N = 26)**

Aligning these elements with the detailed explanations given during the FGDs, six enabling conditions are distilled and will subsequently serve as categories for evaluation using the quantitative data

- Safety from flood risks due to safer location and better dwelling structure and safety from crimes



- Secure housing due to the legal contracts that legitimize their occupancy and assurance of ownership of their homes upon completion of amortizations

- Values of reciprocity among community members and dependable leadership
- Relative stability of income, either through new or expanded livelihood opportunities or reduced cost of living, or expenses on basic services due to subsidies and financial assistance from the local government or other organizations
- Availability of basic amenities, specifically water and electricity; and
- Institutional support, such as capacity building for disaster response and management, financial assistance during

emergencies, and management systems for common service facilities.

**4.3 Reality check: Indicative levels of these enabling factors**

**4.3.1 Safety from flood risk and crime**

Based on the insight of the initial analysis that resettlement has generally increased the safety against natural hazards, this sections delves deeper into the explicit impact of resettlement on the exposure to flooding. This is done by investigating on

the occurrence, the frequency and the occurred damage caused by floods in both, the current resettlement sites and respondent's previous informal settlements. Although the household survey also gathered detailed information on storms, landslides and earthquakes this is not the focus of this study.

When examining the prevalence of flooding, 95,5 % of respondents report no cases of flooding in their current settlement. This marks a significant reduction in flood experience compared to the 8,3 % who did not face flooding in their previous settlement.

This reduction holds true even for the younger resettlement sites, which might have a lower probability of having faced hazards, as the region experienced severe weather events in the past few years, including Typhoons, for example Ulysses in 2020 or Rolly in 2021. Upon further examination in which resettlement sites flooding incidents occurred, it was observed that all 51 cases of flooding were documented in Off-City settlements, representing 6,3 % of the Off-City respondents. These incidents were concentrated in particular locations, namely in Kasiglahan Village, Southville 7 and 8B as well as Theresa LGU. Notably,

Kasiglahan Village and Southville 8B have gained reputation for facing high exposure to flooding in certain areas of the settlements, leading to the perception of residents being "resettled from danger zones to death zones" (Ellao, 2013; Nicolas, 2021).



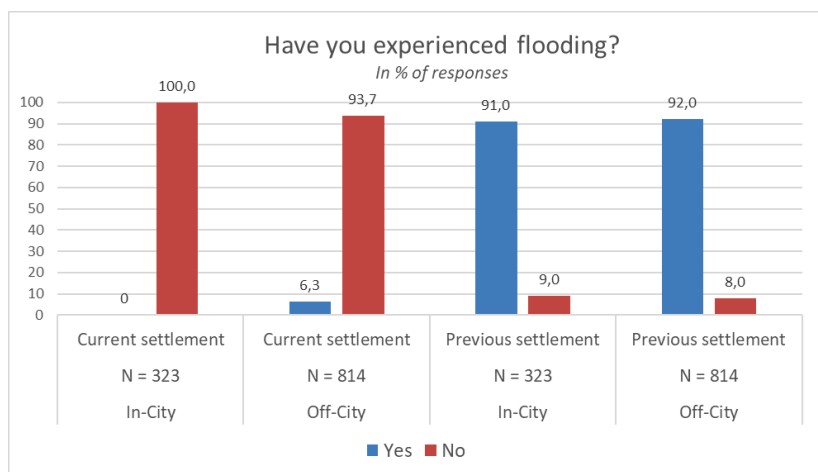

**Figure 6: Comparing flood experiences between resettlement categories**

When respondents who experienced flooding were asked about the extent of damage resulting from these events in their current
        settlement, the majority reported only minor damage. Major damage, such as "collapsed walls" or "houses being washed away"
        were limited to Kasiglahan Village and Southville 7. In stark contrast, the situation was markedly different in their former
        informal settlements locations, where 68,7 % of all respondents reported damage due to flooding. Many of these cases involved
        severe damage and complete loss of household items or even entire dwellings. In response to questions regarding the frequency

of flooding in their previous settlement, 1043 or 79,5 % of the respondents who experienced flooding reported that it occurred
        "several times a year", while 8,6 % indicated that such events occured "every few years" and 10,2 % reported that they
        experienced only a singular flood event.

        The quantitative data aligns with the findings from the FGDs. Participants from Kasiglahan Village recounted their constant
        exposure to floods in their previous homes and shared their constant fear and anxiety every rainy season, especially during

typhoons. The rising water levels of nearby tributaries forced them to evacuate regularly in their former settlement, hindering
        many from going to work. Since their relocation, about 85 % of the FGD participants in Kasiglahan Village feel a sense of
        *"peace of mind"* and *"at ease"*, without worrying about flood water submerging their houses or being forced to take refuge in
        nearby schools. Residents from Saint Therese shared how most of them willingly left their old residences when they were
        offered to relocate to that rather remote place in Rizal province. They mentioned the constant threat of flooding, disruption of

schools, periodic evacuation from their residences to seek refuge in evacuation centers, and dependence on external support
        and subsidies for several days during typhoon season.

        Regarding the threat of petty crimes, the residents claimed that they feel safer, citing the *"peacefulness"* of their current
        neighborhood with community security officers who regularly patrol the area. This observation is particularly reassuring for
        mothers, as it allows children and young people to mingle and play in open spaces. The neighborhood streets are perceived





safer, especially for children, thanks to the enforcement of curfews and notably reduced vehicular traffic within the resettlement area. Also this corresponds with household survey data where 78,5 % of all respondents address that the level of security increased since the move and 16,8 % mentioned it remained the same while only 4,7 % reported about a decrease.

### 4.3.2 Secure housing

The change to a more secure housing in the wake of the resettlement is evident. A notable transition from an illegal status or,
at the very least, an unclear legal status in their former settlements to a legal status is apparent, with 98.9 % of respondents indicating that they did not possess any valid documents, such as contracts or certificates, in their previous settlements. In the resettlement sites, 70,2 % now report having such documents. The availability, or lack thereof, of these documents also influences the perception of secure tenure. Whereas 99,4 % of all respondents reported that they did not have the feeling of secured tenure in the previous settlement, this number decreased to 27,5 % in the resettlement site. Conversely, 72,5 % reported
feeling secure in their tenure. Nevertheless, this also implies that still over one-fourth of resettled households do not have a clarified tenure status or at the very least, do not perceive an improvement in their status. Of particular interest is hereby that the percentage is higher in Off-City settlements with 79,4 %, than in In-City settlements, which report 55,1 %. Upon closer examination, this discrepancy can be attributed a notably sense of insecurity prevalent in Disiplina Village. This is an In-City settlement with a renting scheme, where, other than in all the other resettlement sites, people are not amortizing the house and
or apartment to which they were resettled. Obviously, a significant number of residents in this settlement do not perceive renting as providing long term secured tenure.

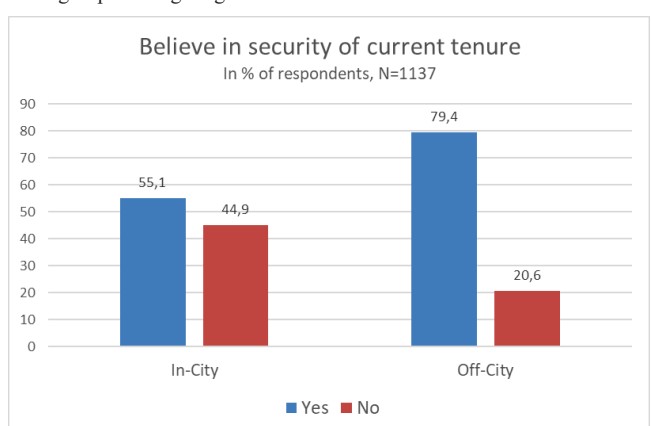

**Figure 7: Comparison of the perception of secured tenure in the current settlement**

The FGDs can provide details on improvement of secure housing but also on why still many do not perceive security. When
asked about the security of the housing, 21 participants agreed that becoming a beneficiary of the housing program and paying monthly amortization increased their confidence in owning their units in the future. *"We can live decently and peacefully*



*because we do not have to worry about flooding in the house and getting evicted or our houses getting demolished".* Others stated that they can now better focus on their work and meet their family's other needs. However, the security of tenure is primarily tied to the capacity to pay the monthly amortization or the rent. Accordingly, the participants from Kasiglahan Village

came to the shared opinion, that the tenure situation remains a source of fear, not providing real housing security. It is reported that many resettles in the site either cannot or do not pay the monthly amortization, creating fear of potential eviction. In this situation, it does not help, that they received the so-called entry pass, stating the right to living in the given house, but not constituting a legal document for ownership.

### 4.3.3 Improved social cohesion

The social relations and networks within the settlement or among the families appear to be well-established in resettlement sites, even slightly surpassing the levels observed in the previous settlements. This is evident when examining the data related to seeking help from neighbors as illustrated in Table 2.

**Table 2: Responses in % to the question: "Would you feel comfortable to ask your neighbours for help (e.g to mind children or support repairing the house), N= 1137**

|              | Current settlement |          |          | Previous   |
|--------------|--------------------|----------|----------|------------|
|              | overall            | In-City  | Off-City | settlement |
| No           | 14,4               | 12,7     | 15,1     | 17,3       |
| I don't know | 1,8                | 1,9      | 1,7      | 1,1        |
| Yes          | 83,8               | 85,4     | 83,2     | 81,6       |


One reason might be that many have family or very close friends in the same settlement, as mentioned by 82,5 % of all respondents. This is only slightly lower than in the original settlement, where 87,5 % mentioned this circumstance. Notably, the percentage is higher for Off-City settlements at 85,3 % compared to in In-City settlements at 75,5 %. This disparity may stem from the limited spaces available in In-City settlements. Another contributing factor is the involvement in self-help and

lending groups, as well as in people's groups or associations such HOAs. Specifically, 83,0 % reported awareness of the existence of self-help groups and lending systems within the resettlement site, which is slightly higher than the 80,3 % in the previous settlement. When asked if they are organized in people's groups or an association, 69,3 % responded "yes", a significantly higher percentage than in the previous settlement where only 29,6 % were organized. The organization rate is higher in In-City settlements standing at 86,7 %, compared to 62,4 % in Off-City settlements, indicating a strong level of

organization within the resettlement sites. Overall, when respondents were asked to rate the solidarity between people in terms of belonging and togetherness in the new settlement, the result was a mean of 7,53 and a median of 8 on a scale ranging from 0 to 10, with 10 as the highest value. The value was slightly higher in Off-City settlements, with a mean of 7,57.



**Table 3: How do you rate the solidarity between people (sense of belonging or togetherness) in the current settlement? Rating from 0 – 10 (10 highest possible), N=1137**

|          | N   | Mean | Median | Std. Deviation |
|----------|-----|------|--------|----------------|
| In-City  | 323 | 7,43 | 8,00   | 2,276          |
| Off-City | 814 | 7,57 | 8,00   | 2,293          |


In the FGDs, all participants affirmed a change in community relations after the resettlement, underscoring their learning from forming a unity and getting organized. This strong form of organization seems to be attributed to the hardship experienced during the resettlement process, where everyone found themselves in a similar situation of starting anew. To overcome this hardship, one opportunity, maybe the only real opportunity, has been to come together, organize each other, helping out and

potentially also articulate the shared interest.

Further, respondents mentioned benefits from having a trustworthy leader and members' cooperation, helping to unify in meeting their basic needs and safeguarding their community from hazards. In Kasiglahan Village, participants shared how their local leader encouraged them to share the cost of a temporary power supply for their houses during the early phase of their resettlement. *"Each household in Phase 1-B regularly contributed five pesos for gasoline to run the association's*

*generator set, which was our temporary source of electricity for three months, while our (association) officers negotiated with the NHA to install permanent supply lines"*. This unity among members empowered the association to advocate for support to install other utilities and amenities, such as water supply and building a local church within the settlement.

In Saint Therese, participants shared that there was no problem among their fellow relocates, but also shared their initial challenges of being rejected by the host community when some barangay residents rejected them upon their arrival. *"Some*

*people used to disturb our site. They threw stones at us or our houses and stole personal belongings like slippers"*. Through collaborative efforts between their association and barangay officers, several assemblies and orientations were held to introduce the relocatees to the initial barangay residents. Over time, the newcomers began joining other community activities, fostering relations with the locals.

### 4.3.3 Income Stability

The household data document an overall reduction of financial capital. The respondents indicated that their income has predominantly decreased since their resettlement, with 48,3 % experiencing a decrease while 18,2 % noted that it remained unchanged and 33,5 % reported an increase in their income. Notably, the decrease in income is significantly higher in Off-City settlements, where 50,9 % reported a decrease, compared to 41,8 % in In-City settlements.



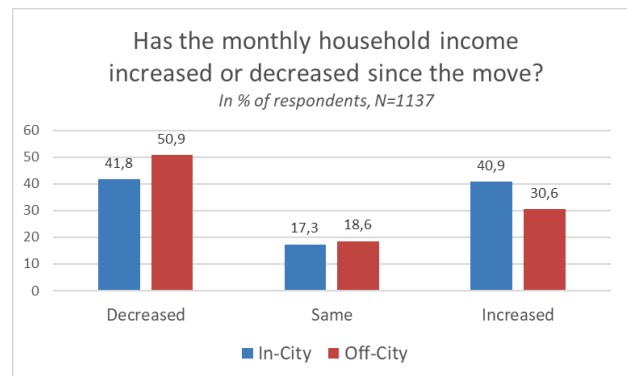

**Figure 8: Comparison of development the monthly income between the two resettlement categories**

The initial analysis of section 4.1 already elaborated that the estimated monthly household income is 16.970 Pesos with a higher mean income of 1.477 Pesos in In-City settlements. When asked if they are able to save money, 71,0 % of all respondents answered "no". This percentage is lower when respondents were asked if they were able to save money before their resettlement, with 53,2 % responding "no". Furthermore, the ability to safe money now is higher in In-City settlements, where 38,7 % claimed to be able to save, compared to 25,2 % in the Off-City counterparts.

The qualitative data from the FGDs show a more nuanced view on the financial situation. Seventeen participants claimed in the discussion that their livelihood slightly improved as income sources became more stable, especially before the COVID-19 pandemic. They explained that this situation did not manifest immediately after moving to the resettlement site. They needed time to adjust and find work. Over the years, most found income opportunities within the site and nearby areas. Some reported diversifying their income sources, providing tricycle service or water delivery. Several residents opened small retail shops, ran home-based food businesses, offered their neighbors housekeeping services, or worked in the local market. Others secured new employment in the nearby areas, leveraging their trade skills for more stable income. However, many also report about people still commuting to urban areas of Metro Manila for work. Often these are husbands or young men, who work for example in the construction sector or as security guards, sometimes only returning to the settlement on weekends. Overall, the FGDs document a general feeling that the incomes have not substantially increased from previous earnings, but for many, the costs of essential services are lower, allowing them to have higher disposable income to cover housing and education needs.

### 4.3.5 Availability of basic services

The access to the analyzed basic services water, electricity and sewage treatment has improved as a result of resettlement. Looking at water supply, 97,8 % now have access to piped water, marking a notably increase from 64,6 % in their former settlement. Access to electricity stands at 99,2 %, with only isolated cases lacking due to various reasons, such as unpaid bills or technical issues. But also in their previous settlements, the access was good, with 93,8 % connected to service providers





and 5 % using so-called jumper connections. In both services, there are marginal differences between resettlement types, with slightly better access reported in In-City settlements (electricity 100 % to 98,9 %; water 99,7 % to 97,1 %). However, access to sewage systems present a different scenario. Two systems are in use: own septic tanks and connection to a public sewage

system. In-City settlements are primarily connected to public sewage system, while in Off-City settlements, the majority relies on septic tanks. In this service category, a substantial improvement is noticeable. In their previous settlements, 41,3 % of respondents reported a lack of sewage system, leading to the discharge of sewage directly to the river or waterbodies.

When regarding the accessibility of education and health facilities, the data indicates mostly similar or better access to schools than before the relocation, whereas for health facilities, the access is reported similar or rather worse. Comparing the two types

of settlements, it is evident that the change in In-City settlements is not that significant, with 47,7 % reporting similar access to schools and 45,5 % for health facilities. However, in Off-City, there is a more noticeable change, showing increased access to schools and a tendency towards relatively reduced access to health facilities.

**Table 4: Responses in % to the task: "Rate the accessibility of human capital facilities now in comparison to your previous settlement", N=1137**

|  | Schools | | Health facilities | |
| --- | --- | --- | --- | --- |
|  | In-City | Off-City | In-City | Off-City |
| Worse | 0,9 | 2,6 | 0,9 | 12,5 |
| Rather worse | 6,5 | 15,2 | 6,2 | 34,5 |
| Same | 47,7 | 35,7 | 45,5 | 28,1 |
| Rather better | 12,4 | 8,8 | 14,9 | 8,6 |
| Better | 32,5 | 37,6 | 32,5 | 16,2 |


The insights from the FGD in Kasiglahan Village suggest that the positive shift in the accessibility of basic amenities documented in the quantitative data only materialized over time and by constant claims. Participants reported that more than 20 years ago, when they first arrived at the settlement, basic services were inadequately provided or even absent. *"In the year 2000, we did not have basic services like school, electricity, water, and church. Water supply came from Pasig City and was*

*delivered to our community at dawn"*. The situation only improved through constant advocacy and claims by HOAs and other civil society organizations. In the younger settlement of Saint Therese such deficiencies in service provision are not reported by the FGD participants. This indicates potential improvements in the resettlement process respectively the implementation of resettlement guidelines and procedures. However, due to the small size of the settlement with only around 250 housing units, respondents noted a lack of public amenities, such as a small public market and a chapel.

**4.3.6 Institutional support to disaster management and of common service facilities**

The survey data provides limited insights into the enabling factor institutional support. Some indicators may partially illuminate this aspect, but not to overall satisfaction. For instance, the questions of whether the resettled individuals received



compensation for leaving their previous settlement is somewhat linked to institutional support. In this regard, 64,3 % of respondents reported receiving compensation payments, with a median value of 18.000 Pesos. This payment serves as initial

support to assist in staring anew, often accompanied by food packages and groceries. Herein, the resettlement of informal settlers, who mostly lack legal documents for their dwellings, is fundamentally different from retreat in formal settings with buyout programs. Another relevant indicator is the provision of livelihood support in the form of training activities. In this context, only 2,9 % reported that they were offered livelihood trainings after the move. Additionally, when asked about the effectiveness of estate management, which exists in all settlements, 75,3 % of respondents reported that it is working fine.

For a more nuanced understanding of this enabling factor, the FGD results, particularly those from Kasiglahan Village, may provide richer insights compared to the quantitative data. These FGD results suggest that the relationship with institutions is not well-established and comes with challenges. This is evident from the absence of basic services in the early years of the settlements and grew alongside the recurring flooding in some parts of the settlement. The residents of Kasiglahan Village were able to address their issues and needs only through the formation of associations and unions among themselves,

demonstrating the importance of bonding social capital. Moreover, the persistent fear of eviction when residents struggle with payments reinforces the view that institutional support is far from perfect.

## 5. Summary of findings

The sequential multi-method research design utilized in this study worked well in addressing the research interest. It facilitated the integration of a comprehensive quantitative dataset with qualitative measures to validate and delve deeper into resettlement

practices. The approach identified community-defined resilience elements, illustrated in Figure 9, that serve as enabling factors for resilient retreat. These elements, at the very least, constitute the minimum preconditions that should be addressed or satisfied by any resettlement initiative. They hold significance as they were not defined as an outcome of literature review or conceptual work of the research team, but by statements of residents when they reflected on community resilience.

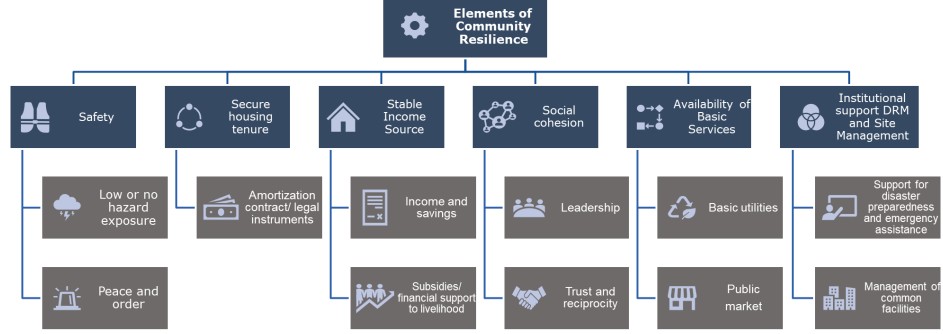


**Figure 9: Emerging model for community defined enabling factors for resilient retreat**



The assessment of these enabling factors across the 10 resettlement sites revealed valuable insights, encapsulated in two overarching findings. Firstly, there is a prevalence of reported improvements in the enabling factors compared to negative trends. That means planned resettlement seems to be more positive than anticipated. Improvements due to resettlement were

particularly related to the reduction of hazard exposure and a higher security from crime. Also basic conditions that may help people to get out of chronic poverty, such as secure land tenure and the access to basic services improved due to relocation in the sites assessed. Also the social cohesion in the new site was positively evaluated. From these factors, particular attention deserve security from hazard impacts and social cohesion. Risk reduction due to a higher security from hazard impacts were primarily achieved by reducing hazard exposure and improved building material (higher robustness). The reduction in hazard

exposure was measured through the three indicators hazard occurrence, frequency and occurred damage, recognizing that the occurred damage can already tend towards risk. It is likely that these indicators cannot completely guarantee hazard security and might not adequately account for future risk associated with further climate change impacting on hazards trends. However, resettlement addressed the most urgent need for exposure reduction. This was crucial for the respondents who faced frequent flooding and damage before their resettlement. Accordingly, the primary objective for the actual resettlement from danger

areas could be achieved for most settlements and for most resettled individuals. This implies a substantial improvement of the precarious living situations of those resettled. However, in resettlement sites where flooding was reported in certain areas, additional measures such as flood protection measures or, in extreme cases, relocation from highly exposed zones may be necessary. The other emphasis is on the strengthened social cohesion as this is countering the risk of marginalization and loss of social structures frequently highlighted in literature on resettlement and displacement. This strengthening can be attributed

to the fact that most resettles did not resettle alone, and that the resettlement process itself served as a catalyst for unity and the formation of associations.

Besides the positive effects on enabling factors, there are negative and inconclusive outcomes for the assessment categories stable income source and institutional support. Concerning monthly income, the data predominantly indicates a decrease. This implies subsequent challenges in other enabling factors, such as housing security. When the income source is unstable or

insufficient, there are not enough financial resources to make the payments for amortization, having the effect that the culture of dependency from the NHA cannot be overcome and, ultimately, eviction from the resettlement site remains a potential threat. The factor institutional support on the other hand, lacks sufficient data. However, insights from the vivid debates of the FGDs suggests ample room for improvement, ensuring that retreat is not only about providing houses and financial schemes for amortizing these houses.

The second major insight of the assessment is that post-relocation conditions only slightly vary between the two resettlement types. The initial analysis (4.1) gave hindsight and the detailed reality check validates these. Accordingly, the results are inconclusive in illustrating that In-City resettlement produces significantly better conditions for the enabling factors than the Off-City modality. On one hand, there are factors where the In-City settlements perform better. Notably, hazard exposure is lower In-City settlements compared to the Off-City category, mainly due to individual Off-City settlements that are affected

by flooding and meanwhile known specifically for high exposure in some parts of the settlements. Additionally, the analysed



income indicators show mainly slightly better outcomes in urban settings. On the other hand, some factors and its individual indicators are assessed slightly higher in Off-City settlements. This is particularly the case for social cohesion. Whereas the picture is less clear for housing security, which largely depends on the different housing types and tenure schemes present in the settlements. In-City settlements predominantly consist of multi-storey medium-rise buildings, while Off-City settlements

are uniformly build with row houses on small lots. It turns out that the renting scheme in the In-City settlement Disiplina Village is perceived providing less secure housing tenure leading to an overall feeling of a less secure tenure in the In-City category.

Hence, the results revealed no discernible pattern of improvement observed in either settlement. This suggests that the location, in the sense of an urban or rather rural setting, may not be a compelling factor for resilience and improved well-being if the

same policies and processes remain in force. This observation is relevant and somewhat contradicts the positions of the ongoing discussion on Off-City versus In-City resettlement. Consequently, the finding that In-City resettlement, in the current practice, might not necessarily be the better option for resettles, needs further argument in section 6 on political implications.

## 6. Conclusion and policy implications

The increasing relevance of retreat as a strategic planning tool for adaptation and risk reduction to extreme events (extreme

storms, floods) and creeping changes, such as sea level rise, underscores the necessity to learn from past and ongoing resettlement experiences. Evaluating what contributes to improving hazard security and living conditions in resettlement sites becomes crucial. The presented enabling factors may serve as an emerging model for assessing successes of resettlement or retreat activities, recognizing the complexity of defining success in such contexts.  This, they may complement more classical analytical models, such as Michael Cernea's IRR model that seeks to identify risks of resettlement projects and developing

strategies to counter them (Cernea, 1997). Positive outcomes in these conditions indicate improvements to the situation before resettlement and can be a pathway to build community capacity to mitigate and recover from hazard-induced disruptions in the future. Resettlement processes can harness the resilience of urban poor communities by prioritizing these conditions when designing specific interventions. It is important to note that the presented approach has limits, particularly in estimating whether an evaluated settlement can fulfil these conditions, such as reducing exposure, in the future, considering potential climate

change impacts. Addressing this aspect would necessitate further research on urban development trends and exposure scenarios.

Contrary to common expectations, the analyzed data reveal that the location of the new settlement, whether in a more urban or rural setting, it is not a major determinant for the success or failure of such a resettlement project. Sustaining a livelihood is challenging in each resettlement site and was challenging in the informal settlements within urban settings. On average,

resettlement neither significantly impoverishes households nor elevates them to a higher livelihood baseline. However, when individuals perceive a reduction in the imminent threat of flooding, there is an opportunity to start anew in a more secure environment with a strengthened community. This suggest that well-planned and managed Off-City resettlement might be a



more viable option than In-City high-rise housing projects. The results further indicate that the first and fourth phases of the resettlement process present entry points for mainstreaming community resilience. Differential vulnerability to climate change
and natural hazards may need to be integrated into the first phase of resettlement, the social preparation of ISFs. These initial and final stages of the resettlement process are evidently critical and could often be the determining factor why projects are not beneficial or at least, why resettlement is often perceived as a negative planning instrument. Retreat needs time (Haasnoot et al., 2021), both for social preparation and the process of starting anew. This insight aligns with classical scientific knowledge on resettlement, such as Thayer Scudder and Elizabeth Colson's (Scudder, 2006; Scudder, 1993; Scudder and Colson, 1982)
and serves as a significant reminder in context of the ongoing agenda for fast and large-scale retreat.

**Data availability.**

Due to privacy issues the questionnaire data are not publicly accessible, but they can be provided through a request to the corresponding author.

**Author contributions.**

Conceptualization: H.L., C.C., E.L., Methodology and software: H.L. and E.L., Formal analysis: H.L., CC. and E.L., and S.I Writing - original draft: H.L. E.L., S.I., Writing - reviewed and edited: H.L. C.C., E.L. and J.B., Project administration: H.L., CC., J.B. All authors have read and agreed to the published version of the paper.

**Competing interests.**

The authors declare that they have no conflict of interest.

**Disclaimer.**

**Acknowledgements.**

This paper presents findings from the research project titled, Linking Disaster Risk Governance and Land Use Planning: the Case of Informal Settlers in Hazard Prone Areas in the Philippines (LIRLAP). This project is funded by the German Federal Ministry for Education and Research (BMBF) under the funding programme "Sustainable Development of Urban Regions",
Grant number 01LE1906C1.

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
