# Peer review of "Risk reduction through managed retreat? Investigating enabling conditions and assessing resettlement effects on community resilience in Metro Manila"

_EGUsphere, 2024_

## Author Comment (AC1)

Dear referee,

thank you very much for your overall positive response to our paper and research. We also appreciate the specific issues you raised and the valuable thoughts. In response, we would like to address your comments.

1. **Please clarify whether retreat, relocation and resettlement are synonyms or define different approaches. If they are considered as synonysms please make use of only one of these terms. Otherwise, please explain the differences.**

Indeed, we are using various terms at different times, partially without defining them in a detailed manner. We commit ourselves to clarify the terms and the use of them in the revised version of the paper. By doing so, we will apply an understanding or definition of the terms retreat, relocation and resettlement in the same way as in our previous work (Lauer et al. 2021). This implies:

- Retreat: "Planned, managed, and permanent movement (retreating) of people and/or infrastructure away from hazard-prone areas to reduce hazard exposure and, ultimately, the hazard risk" (Lauer et al. 2021: 2)
- Relocation and resettlement: Whereas retreat is understood in this study as a mostly scientific and political strategy, resettlement and relocation are the practical components and the terms that are used in practice by stakeholders. Although the terms relocation and resettlement have slightly different meaning, we use them interchangeably in the study (as this is the case in many political agendas in the Philippines). We follow a definition by Fernando (2004) who described both as a "physical movement of people to a new place to live other than the previous place"

*Lauer, H.; Delos Reyes, M.; Birkmann, J. Managed Retreat as Adaptation Option: Investigating Different Resettlement Approaches and Their Impacts—Lessons from Metro Manila. Sustainability 2021, 13, 829. https://doi.org/10.3390/su13020829 )*

*Fernando, N. Forced Relocation After the Indian Ocean Tsunami, 2004: Case Study of Vulnerable Populations in Three Relocation Settlements in Galle, Sri Lanka. Ph.D. Thesis, University of Bonn, Bonn, Germany, 2010.*

2. **A fundamental question is whether the relatively small selection of 3 or 7 case studies for in- and off-citiy resettlement is sufficient to draw general conclusions about the advantages and disadvantages of the two strategies such as " The second major insight of the assessment is that post-relocation conditions only slightly vary between the two resettlement types"**

Certainly, a household survey is inherently limited by its sample size, reproducing "only" a picture or fraction of reality. Due to financial and organizational constraints, we had to restrict the number of settlements investigated. The survey spanned nearly six weeks, and our site selection was carefully undertaken, guided by a detailed resettlement typology developed beforehand (Lauer et al., 2021). This typology considered various settlement aspects, including size, age, housing type, program and involved stakeholders. We aimed to mirror the existing distribution, with most resettlement sites being Off-City sites. Consequently, the majority of our sites are Off-City and "only" three are In-City. But even among these three In-City sites, there is diversity : Bistekville 2 is an LGU-administered site with mixed housing types; Disiplina Village (Ugong) is also LGU-administered but operates under a rent scheme;

Manggahan Residences is an NHA site with a People's Plan approach and medium-rise buildings. With these three settlements, we think we cover a spectrum of settlement types, providing a comprehensive view of In-City settlements.

We appreciate your caution regarding generalizing statements and agree that precision is crucial. To address this concern, we can modify our wording to convey a more nuanced perspective. For instance, we can state: "Based on the obtained data in the selected case study sites, the second major insight of our assessment is that post-relocation conditions only slightly vary between the analysed two resettlement categories."

3. **It is surprising that access to services of general interest (health care, education, culture etc.) was not considered as enabling factor. I can be assumed that in-city resttlement areas perform much better because of their proximity to infrastructure clusters**

Access to the mentioned services was considered as enabling factor, but we named it "Access to basic amenities" respectively "Availability of basic services" in chapter 4.3.5. This chapter delves into the analysis of access to the basic services electricity, water and sewage treatment. Further, we analysed the accessibility of education and health facilities. Notably, we did not assess access to cultural services.

Our collected data indicates a general improvement in the access to electricity, water and sewage treatment across all settlements, with a slightly more favourable outcome in In-City settlements. Furthermore, the access to schools mainly improved in the course of resettlement while the access to health facilities is the only service where there is mainly a noticeable decline in access for those resettled to Off-City location (refer to Figure 4 for the detailed statistics).

4. **The policy recommendations are rather generic. One would like to see more specific recommendations regarding the possible need for amendments to existing legislation and policies for resettlement in the Philippines**

During the research process, numerous findings and policy recommendations were generated and discussed. However, for the sake of shortening the already very extensive paper, these were summarized in the final chapters. Consequently, the recommendations may have become rather generic. In the revised version of the paper, we can commit ourselves checking the recommendations and trying to refine them in order to provide more specific and tailor-made recommendations.

---

## Author Comment (AC2)

Dear referee,

thank you very much for your comment and interest in our research. We appreciate your opinion and input as well as the specific issues you raised and the valuable thoughts. In response, we would like to address your comments.

1. **Introduction: It is not clear what the national water code (1979), based on the danger areas identified, is about. It is mentioned that it does not include flood probability and vulnerability, but what does it include for defining flood risk zones? And how the new settlement locations are identified? Do people know about the flood probability, vulnerability, and exposure of new areas before their relocations?**

The water code does not define flood risk zones. Article 51 of the water code only designates zones near river banks, streams and the shores of the sea and lakes where building structures of any kind are prohibited. In urban areas, this zone extends 3 meters, in agricultural areas 20 meters and in forest areas 40 meters. The water code is the legal basis for justification of no build zones, flood protection and for preventive resettlement.

Subsequent legislation such as for example the Urban Development and Housing Act of 1992 define danger areas allowing eviction and demolition of buildings and structures as following: "When persons or entities occupy danger areas such as esteros, railroad tracks, garbage dumps, riverbanks, shorelines, waterways, and other public places such as sidewalks, roads, parks, and playgrounds".

The landmark decision of the Supreme Court to clear river banks aligns with the water code "Giving urgent dimension to the necessity of removing these illegal structures is Art. 51 of PD 1067 or the Water Code, which prohibits the building of structures within a given length along banks of rivers and other waterways" The major resettlement program "Oplan Likas" then is in line with the water code and the danger area definition.

NHA guidelines (from 2015) exist for site selection and site suitability of resettlement sites. They require site selection outside potential hazard prone and protected areas. However, no clear legislative backing or standardised tools are named. We cannot assure if people who are resettled are informed or aware of potential vulnerability and exposure of the new areas. We can only assume that they trust the argument for relocation, namely that they are resettled from danger areas to safer areas.

The legislation and how resettlement is embedded in strategies in The Philippines is elaborated in chapter 2.2

The named documents are available at these official websites:

- Water Code: https://www.officialgazette.gov.ph/1976/12/31/presidential-decree-no-1067-s-1976/
- Republic Act No. 7279: https://www.officialgazette.gov.ph/1992/03/24/republic-act-no-7279/
- Supreme Court Decision from 2008: https://lawphil.net/judjuris/juri2008/dec2008/gr_171947_2008.html

2. **Figure 1: This figure is confusing: a circular graph, particularly with the arrow connecting the outcome to the start, indicates an iterative process. However, the data collection and analysis process from phases 1 to 3 was a one-off process. The way it is presented in Figure 1 implies that the entire process has been repeated. A linear graph can better explain the process of this study.**

   **Figure 1: If the whole process is connected, with each step building upon the findings of the preceding one, this should be shown by directed arrows indicating the direction of input and output. The type of data transferred between steps can also be shown in these areas, e.g., 'enabling factors for resilient retreat'.**

   **Figure 1 and methodology: I do not understand how the factors identified in the FGD, which is the second step, were already included in the household survey and were then evaluated in phase 3. Is it that the predefined factors identified and included in the HH survey are only ranked (but not identified) in the FGDs? In this case, how is Phase 3 built upon the finding of phase 2? Please elaborate a bit more on the relationships between the three phases and why the analysis needed to be done in separate phases.**

FIGURE 1 will be revised to a linear format to better illustrate the progression of steps and their corresponding input and output data. Based on the newly developed figure, we might also slightly modify the text trying to enhance the clarity in explaining the applied methodology and in how each step builds upon the previous one.

Why was there the need to separate the analysis in three phases? Basically, that structure is an outcome of the work in progress of the whole research process and project proceeding. Meaning that the project planning envisaged the household survey as the first step of primary data gathering. But it was clear that this survey needs to be accompanied by a more qualitative validation. That is why the two FGS were conducted after the initial analysis of the survey. The FGDs not only helped to validate the initial findings but also brought new ideas – the community-defined enabling factors. For us, they were worth to be used as thematic basis for the more detailed analysis of the survey data. This implies that the factors identified in the FGD were not yet included in the household survey. They were identified after the completion of the household survey and thus serve subsequently as a frame to look on the data and analyse it.

3. **The overarching survey questions need to be presented, particularly because the survey and FGD questions are not attached as SM.**

The survey is very long and detailed with 19 pages, more than 70 questions and more than 250 variables making it difficult to name single questions as overarching survey questions. However, the survey was structured into different thematic areas with headlines. These are:

1. Resettlement and mobility profile (Basic questions)
2. Livelihood (Physical, Financial, Human, Social, Natural Capital)
3. Settlement (Hazard profile, Material and design, Planning and comfort)
4. Process (Self-organization, Co-production and participation, Long-term prospect, Governance and trust)
5. Respondent household profile

The FGD was an open and qualitative dialogue following 8 guiding questions.

If wished and needed, we might upload the questionnaire and the guiding question of the FGD as supplementary material.

4. **Line 218: Mai should be May**

Will be changed in the revised version

5. **While graphs are shown for all questions, explaining the content of the graphs in the text is redundant.**

We do not fully understand the comment. We can check the redundancy of explanations and titles of figures and tables with explanations in the text in the revised version of the paper.

6. **Table 2: why an aggregated response on the previous settlement is presented instead of separate in-city and off-city responses to allow for comparison with the current settlement?**

Thank you for that comment. We can delete the "overall" response in table 2.

7. **Limitations: the on-site facilities and services of the new settlements are not included in the analysis. For example, availability or access to basic services might be related to the already available basic services in the new sites rather than their distance from the city. In addition, the period of time since the relocations is also important in what people think about social cohesion or even experiencing flood and there might be a difference between those who moved recently and a long time ago. These kinds of limitations can be acknowledged in the paper.**

A household survey has inherit limitations. Thus, it was crucial to employ a research design, which not only relied on the household survey but also integrated the FGDs as well as various field visits and transect walks. However, still, particularly the diverse nature of the settlements may pose limitations, which is why our site selection was based on a detailed settlement typology. Nevertheless, factors like the age of the settlement, as you mentioned, could influence the outcome. This temporal aspect can potentially influence the likelihood of experiencing hazards since longer-term residency in an area increases the probability of encountering such a hazard in that area. Nonetheless, the results indicate that certain hazards, such as storms and floods, are recurring events happening yearly or even multiple times a year. Moreover, the whole region experienced a series of severe typhoons in the last years, impacting all the surveyed settlements. Accordingly, it is reasonable to assume that the data provides a representative picture of hazard exposure across different sites.

8. **Future study: More granular analysis of relationships among factors and demographic characteristics of the population (age, gender, income, etc.)**

More granular analysis is exactly what is intended to be done in following research work. The dataset is very rich and can deliver more insights, allowing testing different hypothesis and investigate relationships.

---

## Author Response (AR1)

**Comments and Changes**

Dear referees, dear editors,

this document is the point-by-point reply to the comments made by both referees

| **Referee 1** | | |
|---|---|---|
| **No** | **Comment by referee** | **Comment and changes author** |
| 1 | Please clarify whether retreat, relocation and resettlement are synonyms or define different approaches. If they are considered as synonysms please make use of only one of these terms. Otherwise, please explain the differences. | We clarified the use of the terms by defining managed retreat as well as relocation and resettlement in chapter 2.1. In this chapter, we expressed that retreat in this study is understood as the scientific and conceptual strategy while resettlement and relocation are the practical components – mostly used interchangeably. We also newly stated that in the Philippine context, the term resettlement is mostly used (inserted in chapter 1 – introduction).

Following this, we changed and streamlined our wording. Now we only speak of managed retreat as strategy and resettlement as the practical activity. We deleted the term relocation in most instances and use resettlement instead. |
| 2 | A fundamental question is whether the relatively small selection of 3 or 7 case studies  for in- and off-citiy resettlement is sufficient to draw general conclusions about the advantages and disadvantages of the two strategies such as " The second major insight of the assessment is that post-relocation conditions only slightly vary between the two resettlement types" | In our initial reply to the comment, we argued in a detailed manner why we think that the survey provides a comprehensive picture of the settlement types and particularly also the In-City settlements. Nevertheless, we modified the wording to convey a more nuanced perspective to: "Based on the obtained data in the selected case study sites, the second major insight of our assessment is that post-resettlement conditions only slightly vary between the analysed two resettlement categories."

Further: We added the new section "3.5. Limitations" where we acknowledge limitations of a household survey such as the sampling size and settlement selection. |
| 3 | It is surprising that access to services of general interest (health care, education, culture etc.) was not considered as enabling factor. I can be assumed that in-city resttlement areas perform much better because of their proximity to infrastructure clusters | In our initial reply to the comment, we explained that we did consider access to services of general interest as an enabling factor. We named this enabling factor "Access to basic amenities" respectively "Availability of basic services". This enabling factor is explained in chapter 4.3.5.

We hope to satisfy the comment with this reference to the respective chapter. |

| 4 | The policy recommendations are rather generic. One would like to see more specific recommendations regarding the possible need for amendments to existing legislation and policies for resettlement in the Philippines | The final section on "Conclusion and policy implications" was totally restructured and rewritten. A focus is now on providing a conclusion and some specific recommendations including potential entry points for legislative or policy amendments. |
|---|---|---|
| | | However, it we also wish to mention that the major research objective was "evaluating and scrutinizing the suitability of retreat as an adaptation strategy and providing insights into the living conditions of resettled communities". Further, the two concrete research tasks were named: 1) Assess success and 2) Contrast In-City with Off-City settlements. Although the final section is named "Conclusion and recommendations", a research with this scope might not be in the position of providing detailed and fully comprehensive policy or legislative recommendations. This would go beyond what the study promised. |

**Referee 2**

| No | Comment by referee | Comment and changes author |
|---|---|---|
| 1 | Introduction: It is not clear what the national water code (1979), based on the danger areas identified, is about. It is mentioned that it does not include flood probability and vulnerability, but what does it include for defining flood risk zones? And how the new settlement locations are identified? Do people know about the flood probability, vulnerability, and exposure of new areas before their relocations? | We changed the wording in the introduction to: "One concern arises in this context, namely that there is no clarity and consensus on the definition and declaration of danger areas (Republic of the Philippines, 2022). In most cases, they refer only to the national water code from 1976 that defines no-build zones as a buffer of three meters around waterbodies in urban areas. Thus, danger areas are not necessarily areas with a distinct flood probability or taking vulnerable and sensitive elements into account." |
| | | Further modifications and a detailed discussion of the danger area declaration and respective laws in the Philippines seem not appropriate in the introduction to us. But we also now mention in section 6 ("Conclusion and policy implications") the need of establishing a new risk-informed danger area legislation. |
| | | In addition to the changes made in the text, the following elaborations might further provide information to your questions: The water code does not define flood risk zones. Instead, the Code prescribes easements or buffers, among others, between water bodies and various land uses (3 meters for urban areas, 20 m for agriculture, and 40 m for forest use) to protect |

water resources. Only structures relating to navigation, flood management, fisheries, water-dependent utilities, and other installations are allowed in these zones. Residential structures and similar developments are prohibited, which the Supreme Court used in its 2008 decision that directed national agencies and LGUs to rehabilitate the Manila Bay and major river systems and bring the water quality up to code.

The following year, the devastation brought by Typhoon Ketsana (Ondoy) in Metro Manila and neighboring provinces added to the government's resolve to urgently clear waterways and launch a massive resettlement program for the National Capital Region, which the government dubbed as Oplan Likas in 2011.

The government, thus, invoked both the requisite easements in the Water Code and the Urban Development and Housing Act of 1992 (Sections 28 and 29), providing for the humane and participatory resettlement of informal settlers from danger zones in the interest of public safety and protection. The Housing Act defines danger zones as "esteros (creeks), railroad tracks, garbage dumps, riverbanks, shorelines, waterways, and other public places such as sidewalks, roads, parks, and playgrounds." This definition is obviously lacking concrete flood probability or vulnerability and therewith very vague. There are further possibilities by the Department of Environment and Natural Resources (DENR) to define flood risk zones, however also these procedures are lacking a consistent approach and were not the justification for the massive relocation activities happening in the last decades.

The consideration of flood risk and other hazards in selecting housing sites has been part of the housing standards1 developed by the Housing Ministry as early as 1982. This selection parameter remains in force to date. The 2008 guidelines, for instance, explicitly describe the physical suitability of a potential socialized housing site to have
* * *
[1] **Implementing Rules for Batas Pambansa 220**: An Act Authorizing The Ministry Of Human Settlements To Establish And Promulgate Different Levels Of Standards And Technical Requirements For Economic And Socialized Housing Projects In Urban And Rural Areas From Those Provided Under Presidential Decrees Numbered Nine Hundred Fifty-Seven, Twelve Hundred Sixteen, Ten Hundred Ninety-Six And Eleven Hundred Eighty-Five

| | | |
|---|---|---|
| | | "… characteristics assuring healthful, safe and environmentally sound community life. It shall be stable enough to accommodate the foundation load without excessive site works. Critical areas (e.g., areas subject to flooding, landslides, and stress) must be avoided."

Further, NHA guidelines exist since 2015 for site selection and site suitability of resettlement sites. They require site selection outside potential hazard prone and protected areas. However, no clear legislative backing or standardised tools are named.

The research team assumes that the concerned government agencies selected the resettlement sites based on available hazard and risk information and that they involved the concerned communities in the process. Thus, we presume communities trusted the government's assurance of a safer settlement. |
| 2 | Figure 1: This figure is confusing: a circular graph, particularly with the arrow connecting the outcome to the start, indicates an iterative process. However, the data collection and analysis process from phases 1 to 3 was a one-off process. The way it is presented in Figure 1 implies that the entire process has been repeated. A linear graph can better explain the process of this study. | The figure was changed. The new figure is not circular anymore, therewith not implying a repeated process. |
| | Figure 1: If the whole process is connected, with each step building upon the findings of the preceding one, this should be shown by directed arrows indicating the direction of input and output. The type of data transferred between steps can also be shown in these areas, e.g., 'enabling factors for resilient retreat'. | We added information for every different analysis phases which data or findings served as input and what the output of each analysis phase was. We hope that this new figure can resolve the confusion and makes it more easily understandable which data was gathered by which method (survey and FGDs) and how the analysis phases subsequently analysed the data. |
| | Figure 1 and methodology: I do not understand how the factors identified in the FGD, which is the second step, were already included in the household survey and were then evaluated in phase | The enabling factors identified in the FGD were not included in the household survey. They are an output of the FGD respective the following analysis phase. The FGDs were conducted after |

| | | |
|---|---|---|
| | 3. Is it that the predefined factors identified and included in the HH survey are only ranked (but not identified) in the FGDs? In this case, how is Phase 3 built upon the finding of phase 2? Please elaborate a bit more on the relationships between the three phases and why the analysis needed to be done in separate phases. | the household survey – the household survey was the first applied method of data gathering.

Why was there the need to separate the analysis in three phases and to use this multi-stage approach? Basically, that structure is an outcome of the work in progress of the whole research process and project proceeding. The first applied method of the whole research was the household survey. Then, after the first analysis of the household data, it became clear that this survey needs to be accompanied by a qualitative validation in form of the two FGDs. In these FGDs, the participants were asked – among other questions - what factors and which elements are relevant to them for achieving community resiliency in the context of resettlement. In the analysis of these FGD results, we explicitly investigated on this question and distilled the factors the community named and ranked. We found these factors very interesting as they constitute community defined elements and not externally defined ones. We further, found it worth using these factors as a framework for analysing the household survey data and answering if resettlement is improving resilience. Accordingly, we used these factors as framework or categories for analysing the household data gathered in our survey |
| 3 | The overarching survey questions need to be presented, particularly because the survey and FGD questions are not attached as SM. | The different thematic areas of the questionnaire are now mentioned as following in chapter 3.3.:

1. Resettlement and mobility profile (Basic questions)
2. Livelihood (Physical, Financial, Human, Social, Natural Capital)
3. Settlement (Hazard profile, Material and design, Planning and comfort)
4. Process (Self-organization, Co-production and participation, Long-term prospect, Governance and trust)
5. Respondent household profile

We hope this is sufficient. If wished and needed, we might further upload the questionnaire and the guiding question of the FGD as supplementary material, at least we can provide them on request. |
| 4 | Line 218: Mai should be May | Was changed |
| 5 | While graphs are shown for all questions, explaining the content | We were not entirely certain if we understand your comment correctly. Based on our |

| | of the graphs in the text is redundant. | understanding of the comment, we carefully reviewed the graphs and their related explanations in the text. Upon consideration, we believe that retaining the short explanations in the text is the most appropriate approach. The majority of the text's explanations focus on comparing all settlements before and after resettlement. Subsequently, the graphs compare data between In-City and Off-City settlements, offering additional information. We feel that the brief explanations accompanying the graphs are crucial for readers to understand the graphs and their contextual relevance. However, we are open to further discussion and would be willing to consider alternative approaches if necessary. Thank you again for your input, and we appreciate your understanding. |
|---|---|---|
| 6 | Table 2: why an aggregated response on the previous settlement is presented instead of separate in-city and off-city responses to allow for comparison with the current settlement? | The row "overall" was deleted in table 2. |
| 7 | Limitations: the on-site facilities and services of the new settlements are not included in the analysis. For example, availability or access to basic services might be related to the already available basic services in the new sites rather than their distance from the city. In addition, the period of time since the relocations is also important in what people think about social cohesion or even experiencing flood and there might be a difference between those who moved recently and a long time ago. These kinds of limitations can be acknowledged in the paper. | We added the new section "3.5.Limitations" where we acknowledge limitations including the issues you raised. |
| 8 | Future study: More granular analysis of relationships among factors and demographic characteristics of the population (age, gender, income, etc.) | More granular analysis is exactly what is intended to be done in following research work. The dataset is very rich and can deliver more insights, allowing testing different hypothesis and investigate relationships. |